# Structural and Functional Analysis of the Human IQSEC2 S1474Qfs*133 Mutation

**DOI:** 10.3390/biom15050635

**Published:** 2025-04-29

**Authors:** Yonat Israel, Aaron Lowenkamp, Michael Shokhen, Shai Netser, Shlomo Wagner, Joseph Zarowin, Shaun Orth, Veronika Borisov, Orit Lache, Nina S. Levy, Andrew P. Levy

**Affiliations:** 1Faculty of Medicine, Technion Israel Institute of Technology, Haifa 31096, Israel; israel.yonat@campus.technion.ac.il (Y.I.); aaronl@campus.technion.ac.il (A.L.); josephz@campus.technion.ac.il (J.Z.); shaun.orth@campus.technion.ac.il (S.O.); veronikabor@campus.technion.ac.il (V.B.); eorit@technion.ac.il (O.L.); ninal@technion.ac.il (N.S.L.); 2Department of Chemistry, Bar Ilan University, Ramat-Gan 5290002, Israel; michael.shokhen@biu.ac.il; 3Sagol Department of Neurobiology, Faculty of Natural Sciences, University of Haifa, Haifa 3103301, Israel; snetser@univ.haifa.ac.il (S.N.); shlomow@research.haifa.ac.il (S.W.)

**Keywords:** neurodevelopmental delay, conditional mutation, molecular modeling, IQSEC2

## Abstract

IQSEC2 is a guanine nucleotide exchange factor that modulates synaptic transmission, the excitatory/inhibitor balance and memory consolidation. Pathogenic mutations in the *IQSEC2* gene result in epilepsy, cognitive dysfunction and autism spectrum disorder. The most common de novo IQSEC2 mutation in the *IQSEC2* gene, associated with a particularly severe phenotype in males as compared to other IQSEC2 mutations, is due to a frameshift mutation near the C terminus, resulting in an extension of the open reading frame [IQSEC2 S1474Qfs*133]. The objective of this study was to understand the pathophysiology of this specific IQSEC2 mutation using molecular modeling protein–protein interaction assays and a conditional transgenic mouse model of the mutation. Molecular modeling studies showed that the mutation results in the generation of a new domain that may bind ATP. The mutant IQSEC2 protein failed to interact with proteins that normally interact with IQSEC2, most notably with PSD-95. Finally, mice expressing the human mutation displayed marked developmental delays and abnormal social behavior. We conclude that diseases associated with the IQSEC2 S1474Qfs*133 may be due not only to the loss of function of IQSEC2 but also to the appearance of new detrimental activity. The conditional mouse model will allow for the identification of brain regions that are critical for IQSEC2 expression and will serve as a platform for the development of personalized therapies for this disease.

## 1. Introduction

IQSEC2 is a guanine nucleotide exchange factor (GEF) localized to the post-synaptic density of neurons and serves to activate ARF6 by exchanging GDP for GTP bound to ARF6 [1]. The activity of ARF6 has been shown to control AMPA receptor trafficking and dendritic spine maturation. As a result, the IQSEC2 protein plays a major role in controlling synaptic transmission, the excitatory/inhibitory balance and memory consolidation [1].

Pathogenic human mutations in the *IQSEC2* gene result in the triad of drug-resistant epilepsy, cognitive dysfunction and autism spectrum disorder [2]. Children with these mutations generally require 24/7 care. Over 120 different mutations in the *IQSEC2* gene have been identified [1]. The predominant IQSEC2 isoform in the brain is a protein of 1488 amino acids encoded by 15 exons. Approximately two-thirds of these mutations are premature termination codons in exons 1–14 of IQSEC2, which are predicted to result in nonsense-mediated RNA decay with no IQSEC2 protein being produced [1]. The second class of IQSEC2 mutations is missense mutations in single amino acids clustered in specific regions of the *IQSEC2* gene, which encode functional domains of the IQSEC2 protein [1]. Meanwhile, the third functional class of IQSEC2 mutations is due to the generation of a new termination codon in the final exon (exon 15) of IQSEC2 [3]. One such mutation in exon 15 is due to a frameshift mutation in the codon for amino acid 1474 of IQSEC2 and represents a mutational hot spot as it is the single most common site for a de novo IQSEC2 mutation, possibly due to the existence of seven consecutive cytosine residues immediately 5′ to this codon. The most common frameshift mutation at this codon is the insertion of an additional cytosine residue (NM_001111125.2:c.4419dup), resulting in an amino acid change at residue 1474 from serine to glutamine and the generation of a new stop codon 132 amino acids further downstream (denoted S1474Qfs*133) (Figure 1) [1,4].

A major barrier to the development of treatments for IQSEC2 disease is the lack of understanding of how specific mutations affect IQSEC2 function and the absence of disease models of human mutations that may be used as platforms for drug development and discovery. In this study, we sought to understand the pathophysiology of the IQSEC2 S1474Qfs*133 mutation and created a conditional mouse model of this human mutation.

## 2. Materials and Methods

### 2.1. Molecular Modeling of Wild-Type Human IQSEC2 and S1474Qfs*133 IQSEC2 3D Protein Structures

The three-dimensional (3D) protein structure of full-length wild-type human IQSEC2 (1488 amino acid residues) was assembled from the most stable unfolded (US4) conformer of the N-terminal 1094 amino acids of wild-type human IQSEC2 and the carboxy-terminal (C-end) fragment of IQSEC2 (IQSEC2 amino acid residues 1081–1488). We recently reported on the 3D structure of US4 identified in conformational analysis by statistical thermodynamics using accelerated molecular dynamics (aMD) simulations [5]. The 3D structure of the C-end fragment was predicted by means of the DeepFold server. DeepFold [6] is a deep learning-based method for ab initio protein 3D structure prediction starting from a query sequence. DeepFold initially collects multiple sequence alignments (MSAs) from whole- and meta-genome sequence libraries and then generates structural models using an L-BFGS folding algorithm. The US4 and C-end components were joined at their alpha-helical fragments (IQSEC2 amino acid residues 1081–1085) by applying the residue superposition method realized in the YASARA Structure software version 24.4.10 [7,8], after which amino acid residues 1081–1094 of US4 were deleted. Subsequently, the US4 and C-end fragments of IQSEC2 were joined into one composite structure through the formation of a peptide bond between amino acid I1080 of US4 and amino acid A1081 of the C-end segment. Finally, the geometry of the full-length 1488-amino-acid IQSEC2 was optimized with the YASARA Structure software using the ff14SB forcefield [9] in a periodic simulation cell filled with explicit water molecules and Na^+^ and Cl^−^ ions in physiological concentrations at 310 K. The optimized geometry of wild-type full-length human IQSEC2 was used to construct the S1474Qfs*133 mutant of human IQSEC2. S1474Qfs*133 contains 1605 residues, with the mutant fragment modifying the wild-type C-end fragment, starting from residue 1474 and ending at residue 1605 (NM_001111125.2:c.4419dup). In order to model the structure of the S1474Qfs*133 mutant, we first modeled the 3D structure of the S1474Qfs*133 C-end fragment (amino acids 1469–1605) using the DeepFold server. The superposition method was then applied on amino acid residues 1469–1473 of wild-type IQSEC2 and the C-end fragment of S1474Qfs*133. Finally, the optimized geometry of IQSEC2 S1474Qfs*133 was obtained using the YASARA Structure software, as described above.

### 2.2. Molecular Dynamics Relaxation of Wild-Type Human IQSEC2 and Human Mutant IQSEC2 S1474Qfs*133

We performed conventional molecular dynamics (cMD) simulations using the AMBER version 24 software [10] of human wild-type IQSEC2 (1488 residues) and the mutant S1474Qfs*133 IQSEC2 (1605 residues), which had previously been generated via molecular modeling using the YASARA Structure software, as described above. The simulations were conducted using the ff14SB forcefield in a periodic simulation cell with explicit water molecules in a tip3p model with Na^+^ and Cl^−^ ions in physiological concentrations, where the total charge of the system was neutralized by counter ions. After preparing the input topology and structural files, the following steps were conducted before performing the production molecular dynamics simulations: (1) minimizing only the water and restraining the protein (20,000 cycles); (2) a short simulation to allow water to move (NPT, 310 K), restraining the protein; (3) the total minimization of the water and protein (20,000 cycles); (4) molecular dynamics of 1.4 ns to heat the system, restraining the protein (NVT, from 0 to 310 K); (5) relaxing the system and restraining the protein heavy atoms (NPT, 310 K, 1 ns); and (6) relaxing the system (NPT, 310 K, 5 ns). In the equilibration stage, a temperature of 310 K and a stable density were reached. In the final stage (7), we ran short production molecular dynamics simulations at 310 K with a Langevin thermostat and NPT ensemble using pressure of 1 atm and a SHAKE constraint of 2 fs. The production MD run time was 10 ns for both systems in order to relax them.

### 2.3. Functional Analysis of IQSEC2 S1474Qfs*133

We have previously reported on the use of the Lumier assay to quantitatively assess and compare the interaction of wild-type IQSEC2 or mutated IQSEC2 with IQSEC2-interacting proteins [11]. The IQSEC2 mutants used in this study were A350V [11], an N-terminal 213-amino-acid deletion [12] and S1474Qfs*133. In brief, this system uses an N-terminal Renilla luciferase tag of IQSEC2 (wild type or mutant) and a C-terminal triple FLAG tag of the IQSEC2-interacting protein (IQSEC2, apocalmodulin [11] and PSD-95 [13]). pcDNA3.1 vectors using a CMV promoter containing the IQSEC2 and its interactors were transfected into 293 T cells. After the lysis of the cells, the interaction strength was quantitated by the determination of the amount of luciferase activity captured by anti-FLAG antibodies bound to a microtiter plate after normalization to the signal obtained for the total amount of FLAG-tagged protein bound to the plate. The interaction strength was expressed as luciferase/FLAG. Vectors expressing apocalmodulin CALM1 (NM-006888) and PSD-95 (NM_001128827) were obtained from the Human ORFeome Library and subcloned into pcDNA3.1 to create a C-terminal triple FLAG tag [11].

### 2.4. Generation of a Conditional S1474Qfs*133 IQSEC2 Mouse Line

A mouse line for the conditional humanization of the mouse IQSEC2 was generated by Ozgene Pty, Ltd. (Bentley WA, Australia). This model was created by knocking in human exon 15 from S1474 through 107 bp 3′ of exon 15 of the human IQSEC2 gene in place of the corresponding region of the murine IQSEC2 gene via gene targeting in embryonic stem (ES) cells.

A targeting construct was generated to flox murine IQSEC2 exons 13 to 15 with loxP sites and knock in mouse exons 13 and 14 and a chimeric mouse human exon 15 after the floxed region. Chimeric exon 15 consisted of a murine sequence to murine codon P1473 and a human sequence from human codon S1474 onwards. A frameshift was introduced by the insertion of a C nucleotide immediately 5′ of S1474. An additional 107 bp of a human genomic sequence and an exogenous SV40 polyadenylation sequence was also introduced after chimeric exon 15. The targeting construct contained both 5′ and 3′ homology arms that were homologous with the corresponding sequence at the mouse genomic IQSEC2 locus. The 5′ homology arm encompassed IQSEC2 exons 11 and 12 and the 3′ homology arm encompassed the genomic sequence downstream of mouse exon 15. A map of the targeting construct is provided in Appendix A. A double crossover homologous recombination event between the vector homology arms and the genome swapped out the mouse wild-type IQSEC2 exons 13 to 15 and replaced them with the sequence carried by the targeting vector, which encompassed the hygromycin cassette to the neo cassette in **S2**. This included mouse IQSEC2 exons 13 to 15 and then IQSEC2 exons 13 to 15 again, but with exon 15 being the chimeric mouse human exon with the C insertion. The locations of the homology arms are shown in Appendix A (wild-type locus). The targeting construct was fully sequenced by Sanger sequencing prior to electroporation into ES cells.

The targeting construct (Appendix A) was electroporated into male C57BL/6J ES cells (targeted locus shown in Appendix A). Successfully targeted (targeted locus shown in Appendix A) recombinant ES cell clones were initially identified by qPCR and the region of the targeted allele fully sequenced in the area between the selection cassettes prior to being injected into goGermline blastocysts [14]. Male chimeric mice were obtained and crossed to C57BL/6J flp recombinase females to establish heterozygous female germline offspring with the hygromycin and neomycin selection cassettes excised (a map of the conditional allele locus is shown in Appendix A). The flp recombination of the targeted allele excised the selection cassettes shown in Appendix A at the FRT and F3 sites to generate the conditional knock-in allele (Appendix A). Cre-mediated recombination at loxP sites excised murine exons 13–15 to allow for the expression of humanized IQSEC2 with the S1474Q fs*133 mutation (knock-in S1474Qfs*133 allele locus shown in Appendix A) to generate the protein sequence shown in Appendix A. A schematic summary of the breeding strategy used to generate the S1474Qfs*133 mice is presented in Appendix A.

The maintenance and breeding of the IQSEC2 S1474Qfs*133 mice was approved by the Technion Institution Animal Care and Use Committee (IACUC approval # IL-0380322, approval date 3/23/2022). For the maintenance and generation of mice with the conditional knock-in or knock-in alleles, we used the following breeding strategy. Female heterozygous conditional knock-in mice were bred with C57Bl6J males to generate a colony of conditional knock-in mice. Female homozygous conditional knock-in mice were fertile but failed to care for their young. To generate mice with the knock-in allele, female heterozygous conditional knock-in mice were crossed with male mice homozygous for the Cre recombinase (obtained from Ozgene MGI entry https://www.informatics.jax.org/allele/MGI:5435692, accessed on 2 July 2024), with Cre present at the Rosa 26 locus and Cre expression driven by the murine phosphoglycerate kinase (Pgk) promoter. This cross generated male hemizygous mice, female heterozygote knock-in mice and wild-type male and female mice. Only males were used in this study due to the confounding effect of the X inactivation of IQSEC2 in female mice and the lack of X inactivation of IQSEC2 in humans. Knock-in male mice were not fertile. Knock-in female mice were fertile but failed to nurse their pups; thus, it was necessary to maintain the line in a conditional state. The genotyping of the mice was performed by PCR with the primers AACACTTGGCTCACAGTAAGGATG and CTGCACAATGTAGATGATGGTCAG, yielding a 217 bp band for the wild-type allele, a 321 bp band for the conditional S1474Qfs*133 (conKI) allele and a 258 bp band for the S1474Qfs*133 (KI) allele.

### 2.5. Assessment of Ultrasonic Vocalizations in Socially Interacting Mice

Ultrasonic vocalizations were recorded using a condenser ultrasound microphone (Polaroid/CMPA, Avisoft Bioacoustics, Glienicke/Nordbahn, Germany). The microphone was connected to an ultrasound recording interface (UltrasoundGate 116Hme, Avisoft Bioacoustics, Glienicke/Nordbahn, Germany), which was plugged into a computer equipped with the recording software Avisoft Recorder USGH version 4.2.29 (sampling frequency: 250 kHz; FFT length 1024 points; 16-bit format). Ultrasonic vocalizations (USVs) were recorded in 8–10-week-old male wild-type (*n* = 9) or S1474Qfs*133 (*n* = 6) mice during a 5 min interaction with a female C57BL/6 stimulus, following 15 min of habituation to the arena [15]. Ultrasonic vocalizations were analyzed as previously described [15] using our TrackUSF custom-made software version 1.0 (Wagner lab, Haifa, Israel) [16]. MATLAB 2024a was used for statistical analysis. The nonparametric Wilcoxon rank sum test was used to compare the number of USV fragments in the wild-type and S1474Qfs*133 knock-in mice. The raw data for this analysis can be found in Appendix A.

### 2.6. Quantitation of Endogenous Mouse IQSEC2

#### 2.6.1. RNA Extraction and Real-Time PCR

Brains were collected into RNA*later* solution (Invitrogen, AM7020, Waltham, MA, USA). Total RNA was extracted using the Hybrid-R™ Total RNA Isolation Kit (GeneAll Biotechnology, Seoul, Republic of Korea), and DNA was removed using a Clean-Up RNA Concentrator (A&A Biotechnology, Gdansk, Poland). cDNA was obtained using the qPCR cDNA Synthesis Kit (Tamar, PB30.11-10, Mevaseret Zion, Izreal), and real-time PCRs were performed with Fast SYBR Green Master Mix (Applied Biosystems AB-4385612, Waltham, MA, USA). The primers used for the real-time PCRs are listed in Appendix A. The locations of the primers used to quantify IQSEC2 mRNA were between exons 9 and 11 of the IQSEC2 mRNA; therefore, they were located in a region that was common to the wild-type and S1474Q IQSEC2 alleles and outside the region that was genetically manipulated to generate the conKI and KI S1474Q alleles. Murine phosphoglycerate kinase 1 (PGK1) was used as a housekeeping gene for normalization. The real-time PCR program consisted of an initial 20 s at 95 °C and then 30 cycles as follows: 95 °C for 1 s and 60 °C for 20 s. The quantitative real-time-polymerase chain reaction (qRT-PCR) was performed on a StepOnePlus Real-Time PCR System (Thermo Fisher 4376600, Waltham, MA, USA).

#### 2.6.2. Western Blot of Brain IQSEC2

Whole brains were Dounce-homogenized in 250 mM sucrose, 150 mM NaCl, 1 mM EDTA, 1% protein inhibitor cocktail (APE BIO catalog number K1007), 20 mM NaF, 1 mM NaPPi and 1 mM NaVanadate and then centrifuged at 15,000 g for 20 min and the pellet discarded. Then, 40 ug of protein was loaded per well, and proteins were separated on a 7.5% SDS-PAGE gel. Proteins were transferred to a PVDF membrane for 1 h at 100 V in Tris–glycine buffer with 10% methanol. Blocking was performed in 5% skim milk. The primary antibody was GenTex rabbit anti-IQSEC2 (1:1000 dilution, catalog number GTX32143), and the secondary antibody was Jackson Labs peroxidase-conjugated goat anti-rabbit (1:2000 dilution, catalog number 111-035-003).

### 2.7. Statistical Analysis

Data are reported as the mean ± SME, except as indicated for the vocalization analysis, where the median value is shown. The number of biological replicates is indicated for each experiment. Differences between groups were assessed using Student’s *t*-test for all analyses, except for that of the vocalizations, which were not normally distributed, in which case the Mann–Whitney U test was used. A *p*-value of less than 0.05 was considered statistically significant.

## 3. Results

### 3.1. Modeling the Structure of the S1474Qfs*133 IQSEC2 Protein and the Possible Emergence of a New Functional Protein Domain in the Mutant Protein

We modeled the 3D structure of the IQSEC2 full-length wild-type human and S1474Qfs*133 IQSEC2 mutant proteins using the DeepFold server and YASARA, as described in the Methods section. We then used short (10 ns) conventional molecular dynamics simulation via the AMBER version 24 software to generate partially relaxed 3D structures in extended conformational states for the wild-type human IQSEC2 (1488 amino acids) and the mutant IQSEC2 S1474Qfs*133 (1605 amino acids), as shown in Figure 2. DeepFold predicted that the mutation IQSEC2 S1474Qfs*133 would result in the creation of a new ligand-binding site with proposed affinity for adenosine triphosphate (ATP) based on the 3D homology of this site with other proteins, as shown in Appendix A. Figure 3 presents the complex of the carboxy-terminal fragment of S1474Qfs*133 with adenosine triphosphate (ATP) based on the crystal structure of the Mg-ADP-inhibited state of the yeast F1C10-ATP synthase [17] 2wpdA.pdb template (complex 2 from Appendix A).

### 3.2. Steady-State IQSEC2 S1474Qfs*133 Protein Is Decreased in 293T Cells and In Vivo as Compared to Wild-Type IQSEC2

We wished to determine, both in a heterologous system and in vivo, whether there were differences in the steady-state amount of IQSEC2 protein as a result of the S1474Qfs*133 mutation. The transfection of luciferase-tagged wild-type IQSEC2 vs. S1474Qfs*133 IQSEC2 into 293T demonstrated an approximately 75% reduction in the amount of total luciferase protein in the cell extracts (77 ± 8%, *n* = 3 independent experiments) of luciferase S1474Qfs*133 IQSEC2-transfected cells. This suggests that the luciferase-tagged S1474Qfs*133 may be either translated less efficiently or be less stable than the wild-type IQSEC2 protein in this heterologous system. As assessed by the Western blot on the whole brain, there was an approximately 20% reduction in the amount of IQSEC2 protein in S1474Qfs*133 mice compared to wild-type mice (Appendix A), with no significant difference in brain IQSEC2 mRNA between the S1474Qfs*133 and wild-type mice (Appendix A), suggesting that the reduced steady-state amount of the S1474Qfs*133 protein in the brain may be due to differences in the translation of the mutant RNA or the stability of the mutant protein.

### 3.3. Impaired Protein Interactions of the S1474Qfs*133 Protein

We sought to compare the interaction strength of the wild-type and S1474Qfs*133 IQSEC2 proteins with known interactors of IQSEC2 using the Lumier system, as described in the Methods. Figure 4 demonstrates the interaction strength (expressed as luciferase/FLAG) between the wild type and the S1474Q mutant and several interacting proteins [11]. We anticipated that the binding of PSD-95 to the mutant S1474Qfs*133 would be eliminated as the PSD-95 binding site (amino acids 1484–1488 of wild-type IQSEC2) is destroyed by the S1474Q fs*133 mutation [1,3,13], and we indeed found a complete lack of binding of the mutant (99 ± 0.2% reduction in binding). We also observed that the S1474Qfs*133 protein had impaired interactions with apocalmodulin and an impaired ability to form an IQSEC2–IQSEC2 dimer.

₋**Panel (A):** Interaction strength between luciferase-tagged IQSEC2 and FLAG-tagged apocalmodulin [11]. WT—wild type IQSEC2. A350V—missense mutation of IQSEC2 in the IQ domain, previously demonstrated to impair binding of IQSEC2 to apocalmodulin [11]. N del—deletion of the N-terminal 213 amino acids of IQSEC2 [12]. S1474Qfs*133 mutant. There was an 89 ± 1% reduction in binding of the S1474Qfs*133 mutant to apocalmodulin as compared to WT IQSEC2 (*n* = 8 for each interacting pair, *p* < 0.01 between wild type and S1474Q).₋**Panel (B):** Interaction strength between luciferase-tagged IQSEC2 and FLAG-tagged IQSEC2, assessing dimerization [11]. IQSEC2 has been demonstrated to dimerize as mediated by its N-terminal 213 amino acids [12]. Dimerization was assessed using the Lumier method [11]. There was an 87 ± 6% reduction in the ability of the S1474Qfs*133 mutant to dimerize as compared to WT IQSEC2 (*n* = 8 for each interacting pair, *p* < 0.01 between wild type and S1474Q).₋**Panel (C):** Interaction strength between luciferase-tagged IQSEC2 and FLAG-tagged PSD-95 [11]. The interaction between IQSEC2 and PSD-95 is affected by the PDZ domain of IQSEC2, corresponding to the carboxy-terminal 4 amino acids of IQSEC2 [13]. The S1474Qfs*133 mutant had a 99 ± 0.1% reduction in binding to PSD-95 as compared to wild-type IQSEC2 (*n* = 8 for each interacting pair, *p* < 0.01 between wild type and S1474Q).

### 3.4. Early Mortality and Growth Retardation of S1474Qfs*133 Mice

We anticipated that, in male mice, the S1474Qfs*133 mutation would result in early mortality and delayed weight gain as compared to wild-type mice, as male children with the mutation have a severe delay in their growth and multiple feeding problems. We refrained from counting mouse pups within the first 24 h after birth as we observed that this caused the cannibalization of the pups. At 24–48 h after birth, there were consistently two to three dead mice found in each litter, and there was skewing in the genotypes that were taken from the living mice at 24–48 h after birth. We expected 50% of the male pups to be wild types and 50% to be knock-ins. However, we observed a 2:1 ratio of wild-type males at 24–48 h after birth (32 vs. 16 in 14 litters). We inferred that approximately 50% of the male knock-in mice died within the first 24–48 h after birth. We continued to follow the growth of wild-type male and mutant male littermates. Figure 5 shows that there was a highly significant reduction in weight in the male S1474Qfs*133 male mice as compared to their wild-type male littermates (N = 3–6 mice at each time point for each genotype measured through 20 days of age).

### 3.5. Impaired Ultrasonic Vocalizations in Socially Interacting S1474Qfs*133 Mice

Male mice are known to emit ultrasonic vocalizations (USVs) when introduced to female mice (mating calls) [15]. We recorded the mating calls (Figure 6A) of IQSEC2 wild-type and S1474Qfs*133 littermates and analyzed them using our custom-made analysis system, TrackUSF, which enables the separation of ultrasonic vocalization fragments (USFs) from noise signals and the quantification of the USFs [16]. We found that, while all wild-type animals (*n* = 9) emitted ultrasonic vocalizations (Figure 6B), none of their S1474Qfs*133 littermates (*n* = 6) exhibited such behavior, and this difference was statistically significant (Figure 6C; Mann–Whitney U test = 99, *p* < 0.001). These results are similar to the results obtained with A350V IQSEC2 male mice [15] using the same method.

## 4. Discussion

We have provided here a plausible mechanism for the pathophysiology associated with the S1474Qfs*133 mutation, which involves both a partial loss of the normal function of IQSEC2 and the possible creation of detrimental activity as a result of the formation of an ATP-binding domain. The mutation results in a change in the last four amino acids of IQSEC2 at the C terminus of the protein, as well as a significant C-terminal extension of the protein. In the wild-type IQSEC2 protein, the last four amino acids STVV represent a consensus sequence for the binding to type I PDZ domain proteins [18]. The major PDZ protein with which IQSEC2 has been shown to interact is PSD-95, which serves as the primary scaffolding protein at the post-synaptic density (PSD). We have shown here that S1474Qfs*133 fails to bind to PSD-95. Previous work has established that the failure of IQSEC2 to bind PSD-95 results in a change in the intracellular distribution of IQSEC2 from being located at the PSD in the dendritic spine to being diffusely distributed along the dendritic spine [3]. As IQSEC2 has been shown to regulate AMPA receptor trafficking [12,18], the disruption of the localization of IQSEC2 would be anticipated to result in abnormal AMPA-mediated responses. This in turn would be expected to have a major impact on the formation of new dendritic spines [19] and on the excitatory/inhibitory balance. Taken together with the reduction in the amount of total IQSEC2 protein resulting from this mutation, it is anticipated there will be a significant decrease in the GEF-dependent and GEF-independent functions of IQSEC2. We have also identified a possible harmful affect of this mutation as a result of the formation of a new ATP-binding domain, which might be expected to interfere with the energy-dependent processes in the neuron.

The S1474Q male mice displayed early mortality, a marked impairment in weight gain and the complete absence of female-stimulated vocalizations, which were more severe than in other IQSEC2 mutant mouse lines that have been studied [11,20,21]. This is consistent with the clinical course of male children with the S1474Q*fs133 mutation, who have a more profound growth and neurodevelopmental delay than children with other IQSEC2 mutations [4]. In particular, male children with this mutation suffer from marked feeding issues (i.e., reflux and dysphagia) and hypotonia [4]. The increased severity of this mutation may be due to the gain-in-function mutation involving ATP binding, as discussed above. Future studies will address whether inhibiting the binding of ATP to S1474Qfs*133 mice may have a benefit.

We created the S1474Q model in a conditional format, anticipating that the S1474Q mice may have reduced viability and fertility, which was borne out in our development of the colony. Importantly, this represents the first conditional IQSEC2 mutant mouse line. This line will be instrumental in determining which brain regions are critical for the normal functioning of IQSEC2 by crossing mice with the conditional mutation with mice expressing CRE recombinase under the control of a brain region-specific promoter. For example, a CRE line under the control of a hippocampal promoter may be used to selectively mutate IQSEC2 in the hippocampus.

This study has several limitations. The potential ability of the C terminus of the IQSEC2 S1474Qfs*133 mutant protein to bind ATP has only been identified using homology modeling and has not been assessed experimentally. We inferred that there was reduced survival in mutant vs. WT male mice from the skewed distribution of WT vs. mutant male mice when we genotyped the mice on day 2 (as presented in the Results). The reduced survival of the mutants appeared to occur in pups within 24–48 h after birth, as evidenced by the finding of dead pups in this period. We were not able to genotype the mice immediately after birth, as the handling of the mice at this time resulted in the mother cannibalizing all of the pups. We cannot rule out that mutant mice may have increased mortality in utero, and this could be addressed through the genotyping of embryos at different stages of development. After the 24–48 h post-birth period, the mutant mice survived, although their growth was markedly stunted, as shown in Figure 5. Finally, we have demonstrated altered behavioral social interactions singly based on male and female vocalizations, but it may be possible to obtain a stronger conclusion regarding impaired social behavior using additional behavioral tests, such as the social preference/novel mouse paradigms.

The data presented here suggest that diseases associated with the IQSEC2 S1474Q*fs133 mutation may be due to not only the loss of normal function of IQSEC2 but also to the creation of detrimental activity. We therefore propose that therapies for this mutation may require a combined knockdown and replacement approach—specifically, the knockdown of the mutant protein mediating the aberrant function (using short hairpin (shRNA) or antisense oligonucleotides to promote mRNA degradation) together with a functional IQSEC2 gene delivered to the neuron by an adeno-associated virus (AAV).

## 5. Conclusions

The IQSEC2 S1474Q*fs133 mutation likely results in an impairment in the ability of IQSEC2 to localize properly to the post-synaptic density due to the loss of its PDZ-binding domain [13,18], where it normally mediates its GEF function. Molecular modeling studies show that the frameshift mutation results in the generation of a new ATP-binding domain that may be deleterious by binding and sequestering ATP. We hypothesize that diseases associated with the IQSEC2 S1474Qfs*133 mutation may require treatment with a knockdown and replacement approach to achieve efficacy.

## Figures and Tables

**Figure 1 biomolecules-15-00635-f001:**
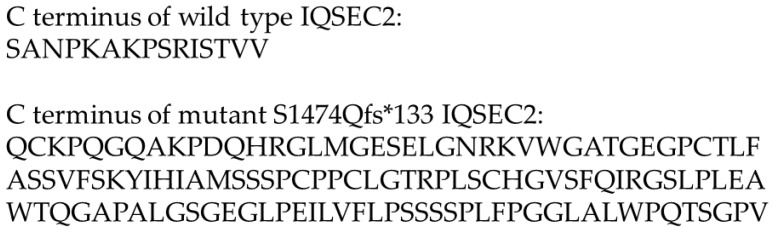
Carboxy (C)-terminal amino acid sequence of the human wild-type IQSEC2 protein and the human mutant IQSEC2 S1474Qfs*133 protein. The sequences shown start from amino acid 1474 (mutant and wild type are identical prior to amino acid 1474). The complete amino acid sequence of the human IQSEC2 S1474Qfs*133 protein is provided in Appendix A.

**Figure 2 biomolecules-15-00635-f002:**
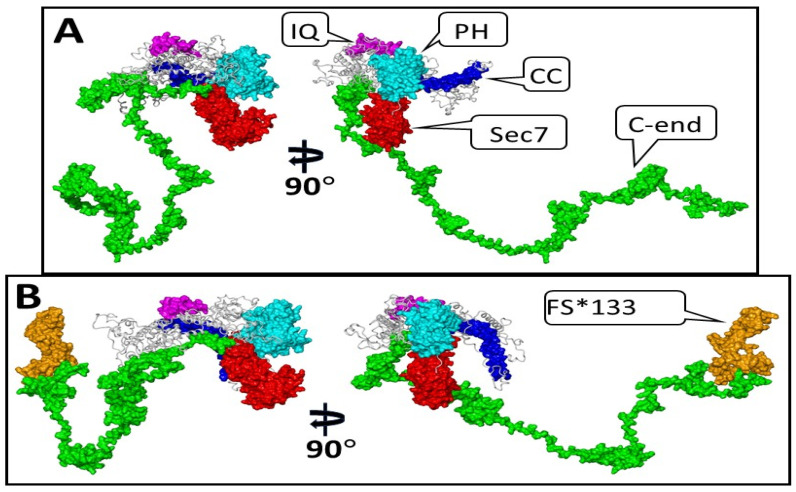
Three-dimensional structures of wild-type human IQSEC2 and S1474Qfs*133 IQSEC2 in two projections. The functional domains are presented as their molecular surfaces. Panel (**A**)—IQSEC2 wild type. Panel (**B**)—S1474Qfs*133 IQSEC2 mutant. Color scheme of different domains and amino acid residues corresponding to these domains: CC (coiled coil) domain aa 23–74, blue; IQ (apocalmodulin binding) domain aa 347–376, magenta; Sec7 (GEF catalytic domain) aa 746–939, red; PH (plextrin homology) domain aa 951–1085, cyan; C-end (c terminal fragment aa 1095–1488 for panel (**A**) and aa 1095–1473 for panel (**B**), green; S1474Qfs*133 (132 aa added due to frameshift mutation at C-terminus of S1474Qfs*133 as in panel (**B**)) aa 1474–1605, orange; the remaining fragments of IQSEC2 are colored in grey and are presented in a ribbon style.

**Figure 3 biomolecules-15-00635-f003:**
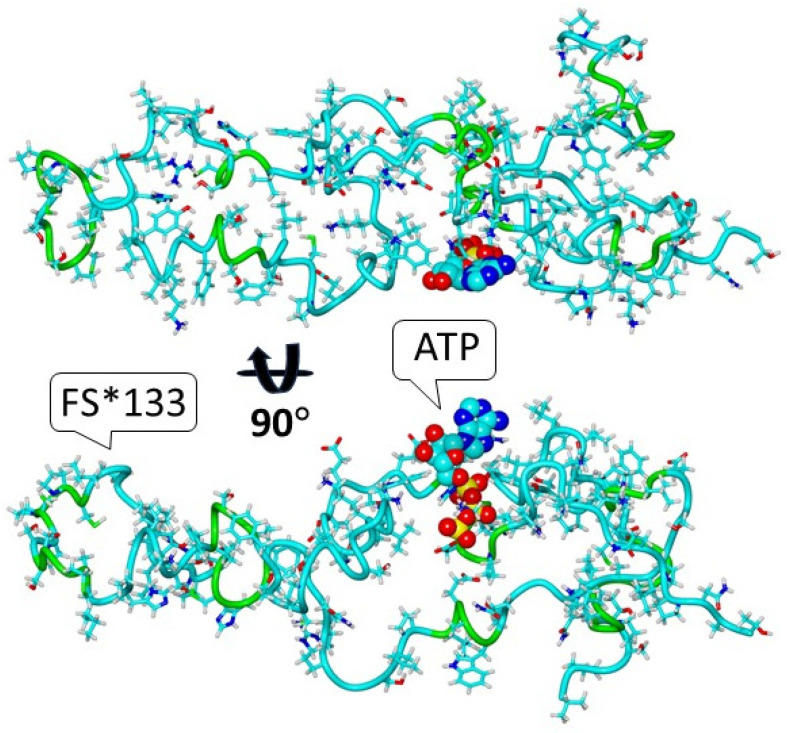
The complex predicted by DeepFold of S1474Qfs*133 with adenosine triphosphate (ATP) in two projections.

**Figure 4 biomolecules-15-00635-f004:**
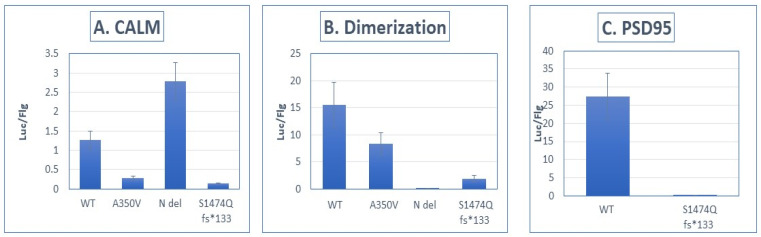
Interaction between S1474Qfs*133 and PSD-95 and other IQSEC2-interacting proteins is impaired.

**Figure 5 biomolecules-15-00635-f005:**
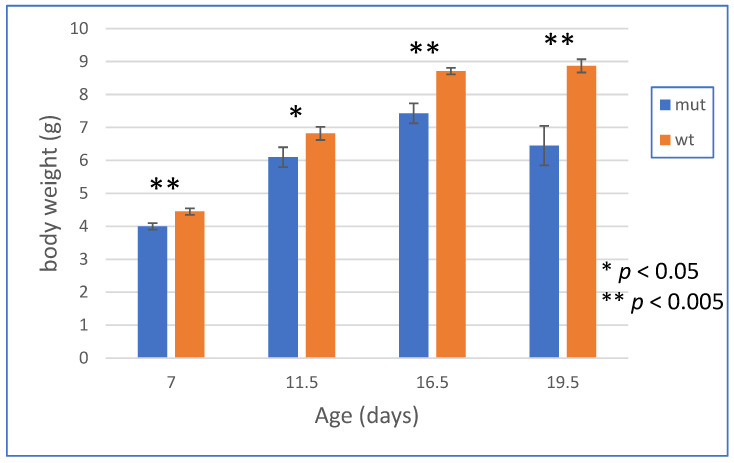
Growth chart of WT and S1474Qfs*133 mice.

**Figure 6 biomolecules-15-00635-f006:**
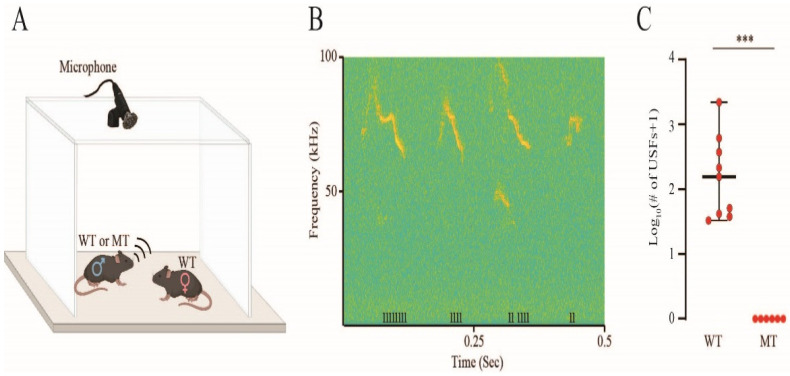
IQSEC2 S1474Qfs*133 mutant male mice do not emit mating calls. (**A**) Schematic depiction of the recording arena equipped with an ultrasonic microphone. Wild-type (WT) or mutant (MT) male mice were exposed to a wild-type female for five minutes in this arena. (**B**) A representative spectrogram showing the ultrasonic audio signals recorded during a 500-millisecond segment of the 5 min exposure of a wild-type male to a female. Ultrasonic fragments (USFs) detected by the TrackUSF analysis system are denoted at the bottom of the figure by the notation “I”. (**C**) Comparison of the number of USFs between wild-type and mutant IQSEC2S1474Qfs*133 male mice. The Y-axis shows the Log10 transformation of the number of USFs (+1). Mutant mice did not emit any vocalizations, while wild-type mice vocalized as expected (Mann–Whitney U test. U = 99, *** *p* < 0.001). The median, maximum and minimum USF values are indicated with a horizontal bar.

## Data Availability

All research data can be obtained by contacting the corresponding author.

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
