# Peer review of "Structural and Functional Analysis of the Human IQSEC2 S1474Qfs*133 Mutation"

_biomolecules, 2025, doi:10.3390/biom15050635_

Round 1
Reviewer 1 Report
Comments and Suggestions for Authors
The abstract must be restructured to enhance clarity. What is the main goal of this study? What is IQSEC2? Is it a gene? Protein? Transcription factor? The abstract needs to be restructured for improved clarity. What is the main goal of this study? Is IQSEC2 a gene, a protein, or a transcription factor? Additionally, the readability and comprehension of the abstract should be enhanced. The methodologies applied in the study should also be detailed. After reviewing the abstract, I was unable to identify the main findings. Please revise this section.
While the introduction is intriguing, achieving a more technical tone requires refinement. In the objectives, the authors should specify the population on which the study was conducted. Figure S1 could be included in the main document due to its relevance to understanding the study.
No description was provided regarding the statistical evaluation. It is essential to report data normality and the statistical tests applied.
Has the novel ATP-binding domain been experimentally confirmed to bind ATP? If so, under what conditions?
What structural elements in the new C-terminal extension might mediate its binding to ATP or other cofactors?
Could the new domain interact with additional synaptic proteins not screened in this study?
Could RNA-based therapies (e.g., antisense oligonucleotides) provide alternative means to target the mutant transcript? Please better explore such a topic in the discussion section.
Are specific brain regions or cell types where the mutant protein accumulates or causes more pronounced effects?
What are the implications of this mutation for heterozygous female carriers—was mosaic expression modeled or analyzed?
Is the gain-of-function effect due solely to ATP sequestration, or might the mutant interfere with endogenous protein networks?
How does the severity of this mutation compare with truncating mutations that result in complete loss of IQSEC2 expression?
Overall, the study is engaging, though it appears preliminary. The implications of the mutation and potential in vivo findings should be discussed in greater depth.
What are the potential limitations of this study? It must be recognized to enhance data interpretation. The perspectives must also be mentioned.
Lastly, the conclusion section is too long and should be reduced.
Comments on the Quality of English LanguageThe document could be polished regarding English grammar and syntax.
Reviewer 2 Report
Comments and Suggestions for Authors
In this paper by Israel et al, Authors discuss a new IQSEC2 mutation that causes a potentially detrimental gain of function at the C-terminal domain with concurrent loss of normal IQSEC2 binding properties.
I have some minor comments regarding the formatting of the manuscript: font/size seem different on lines 86-95; there are small typos and missing spaces (i.e. line 114). It would also be helpful if Authors would explicitly mention the acronyms the first time they are encountered in the text (i.e. ES cells). Reference 16 looks a different color from the rest of the text. Also I'd rename supplementary table 2 as 3 and viceversa as reference to table 3 is encountered earlier than the current table 2 in the text. There is amissing period in line 216 and the italic formatting is missing in line 218. Also the formatting of the Results section is different (margins). I would encourage the Authors to use a consistent verb tense throughout the manuscript, when possible (i.e. line 235 switches from past to present). It appears that lines 250-1 are a duplicate of 253-4. Please revise. Line 280 could also use some formatting adjustment. Line 272 would read better as "known interactors OF IQSEC2". Line 314 - please spell "thru" correctly. Quality of the graph in figure 4 is quite low - I'd encourage the Authors to either use a different software or improve the quality of the existing image. Finally, please consider adding spaces starting at line 398/399 and subsequent new paragraphs.
I also have some minor and major comments regarding the study/research:
MINOR - 1) the breeding strategy described in the M&M section could be made easier with a graphical schematic as a supplementary or main figure. 2) The Authors mention experiments performed on 293T cells but I am unable to locate them in the figures as well as there is no reference for them. Please add these results in a main figure. 3) Discussion could be expanded with more comparative results to previous mouse models or clinical data.
MAJOR - 1) It would be interesting to see data plotted for reduced survival of the mutants vs WT mice. Has this been evaluated? 2) Are there any brain phenotypes or malformations? A morphological brain comparison could be added in a supplementary figure. 3) It is mentioned altered socializing behavior only based on mating vocalizations. As interesting as this result is, to make a stronger conclusion on impaired socialization it would be good to include at least another experiment, such as novel object/novel mouse trial.
Round 2
Reviewer 1 Report
Comments and Suggestions for Authors
The authors adequately addressed to all my concerns. I thank you the consideration. I have no further comments.
Reviewer 2 Report
Comments and Suggestions for Authors
Thank you for addressing my comments.
-Line 180: please correct the spelling of "embryonic".
-It might be helpful to add a reference to figure 4 in the section described at lines 329-330 for the 293T experiments to make it clear.
-The answer provided to my question about reduced survival could be a great addition to the text - please consider adding this in the discussion or when referencing to fig. 5.
-The comment about adding a limitation to the study section was not addressed. Considering some of the answers (i.e. need for more behavioral testing, limitation in not being able to genotype early) and my previous comments, these should be addressed in the manuscript. If the Authors would prefer to not add a separate section, please consider addressing these comments/limitations in the discussion or conclusion sections.
